# Use of Stem Implanted Bioherbicide Capsules to Manage an Infestation of *Parkinsonia aculeata* in Northern Australia

**DOI:** 10.3390/plants10091909

**Published:** 2021-09-14

**Authors:** Victor J. Galea

**Affiliations:** School of Agriculture & Food Sciences, The University of Queensland, Gatton, QLD 4343, Australia; v.galea@uq.edu.au; Tel.: +61-7-5460-1282

**Keywords:** stem implanted capsule, bioherbicide, parkinsonia, woody weed, dieback, mechanical delivery

## Abstract

An infestation of parkinsonia (*Parkinsonia aculeata*) located on Alexandria Station, Northern Territory, Australia, was successfully treated with a bioherbicide using stem-implanted capsules. The bioherbicide (Di-Bak Parkinsonia^®^), containing three endemic endophytic fungi (*Lasiodiplodia pseudotheobromae*, *Macrophomina phaseolina* and *Neoscytalidium novaehollandiae*), is the first Australian registered woody weed bioherbicide. The product was effectively administered to the plant stems using a mechanical device, resulting in the subsequent development of a dieback event. After a period of establishment, it progressed through an adjacent untreated population, resulting in a significant decline in infestation vigour and preventing recruitment from the seedbank. This is the first report of large-scale management of parkinsonia by this method.

## 1. Introduction

Parkinsonia (*Parkinsonia aculeata* L.), also known as Mexican palo, Jerusalem thorn, blue palo verde, horse bean tree, sessaban and Barbados flower fence [1], is a thorny shrub that can grow into a small tree with a natural range from the southern USA through to Argentina [2]. Parkinsonia is thought to have been introduced in Australia in the 1860s [3], quickly spreading through its utility as an ornamental shade tree and, by 1906, was considered a weed [4]. Parkinsonia populations are greatest in northern Australia (Queensland, Northern Territory and Western Australia), with only a few scattered infestations reported in New South Wales and even fewer in South Australia. It is estimated to be present in over 3.3 million ha of Australia [3] and has been classified, in this country, as a Weed of National Significance (WoNS) in recognition of it being a major weed threat. This classification identifies parkinsonia for implementation of strategic action plans (at various jurisdictional levels) and as a priority species for management programs and targeted research, as it produces impenetrable thickets which impact grazing systems [5] and natural riparian zones.

Dieback of parkinsonia is a documented phenomenon leading to plant mortality, and sometimes entire populations are affected [5]. The first detailed examination of parkinsonia dieback across northern Australia [6] resulted in the identification of 41 fungal species from 13 families. Among these, eight species from the family Botryosphaeriaceae were found to be common across all five climatic regions sampled in dieback-affected parkinsonia. A seven-year study of a naturally occurring dieback event near Hughenden, Queensland [6,7] demonstrated the linear movement of this disorder through a parkinsonia population, killing established plants and preventing recruitment of juveniles. An inoculation methodology, progressively developed in [6], proved successful in establishing dieback within healthy individuals using French white millet (grain) colonised with certain species selected from the fungal bank. Subsequent research developed an advanced process [8] in which colonised millet was formulated in either gelatin or hypromellose pharmaceutical capsules which has been utilised by other members of this research group [9,10,11]. This technology, based on Australian patent # AU 2009201231 B2, was further developed into a mechanised delivery system [12], which was used in this research.

A recent comprehensive review on biological control of weeds with plant pathogens [13] identified that only 15 bioherbicides (13 based on fungi) are registered for use in various countries; of these, only two (Di-Bak Parkinsonia^®^ and SolviNix™), are currently commercially available. Among the discontinued products, Chontrol™ and Myco-Tech™ (based on *Chondrostereum purpureum*) and Stumpout^®^ (*Cylindrobasidium laeve*) are the only products specifically developed for the management of woody weeds. The products based on *C. purpureum* were registered for control of plants in the genera *Acer*, *Alnus*, *Betula*, *Populus* and *Prunus*, in North America, while *C. laeve* was active against *Acacia mearnsii* and *A. pycantha* in South Africa. Typically, products containing *C. purpureum* are applied as a paste, containing mycelium, to freshly cut stems of the target weed. Colonisation of the stump by this pathogen/saprophyte prevents re-sprouting by blocking the vascular system [14]. The product, based on *C. leave,* is formulated as an oil suspension that is painted onto cut stump surfaces where the saprophytic fungal inoculant causes decomposition [15]. 

Glasshouse studies by another research group [16] used their own isolates of endophytic fungi from both healthy and dieback-affected parkinsonia populations near Charters Towers (north Queensland) and a benchmark isolate (*Lasiodiplodia pseudotheobromae—*NT039) selected from this laboratory [6,17,18]. These researchers failed to produce dieback symptoms or systemic infection in juvenile parkinsonia plants under three water stress regimes.

Although parkinsonia dieback-affected plants may be associated with a variety of endophytic fungi, possibly as latent pathogens, the authors [16] suggested that the syndrome itself may require an external environmental trigger to initiate disease, which implies that inoculation alone may not be sufficient to induce disease in the field.

The current study aimed to evaluate the effectiveness of a stem implanted capsule formulated fungal bioherbicide for the control of a vigorous population of parkinsonia growing naturally in a rangeland situation. This study also reports the first large scale use of a mechanised delivery system to enable rapid and effective inoculation of parkinsonia and the establishment of a successful dieback control event.

## 2. Results

### 2.1. Weather Data

Data for the site (Figure 1a) indicate that in the 12 months following the establishment of the trial (August 2016–August 2017), rainfall was within the expected range; however, the following 12 months were unusually dry with the failure of the 2017–2018 wet season. Similarly, there was less than average rainfall in the 2018–2019 season, with a return to expected levels for the 2019–2020 season. Temperature readings from Camooweal over the period of the trial (Figure 1b) indicate that the monthly mean minima ranged between 8.5 and 26.9 °C, while the monthly mean maxima ranged between 25.7 and 42.0 °C.

### 2.2. Dieback Development in the North Infestation 

Development of dieback in the north infestation was incremental and significantly greater (*p* < 0.01) than that observed in the untreated control areas, as evidenced by the steadily falling health scores for all assessment dates after treatment (Figure 2). Health had fallen to a rating of 3.82 by 64 weeks after treatment (WAT), then to 1.18 and 0.35 by 165 and 221 WAT. There was a comparatively smaller reduction in health in untreated plots falling from 5.0 to 4.34 and 4.36 by 165 and 221 weeks, respectively. Although these reductions in health were significant (*p* < 0.01), they are minimal and were expected as a consequence of an ageing plant population. 

Mortality assessment (Figure 3) indicates that falling health scores rapidly translated into dead parkinsonia plants. At 45, 64, 165 and 221 WAT mortality averaged 6.0, 9.0, 55.3 and 81.1% with each increment significantly (*p* < 0.01) greater than the previous. Visually, plant mortality was evident (Figure 4) with treated trees mostly dead, in some cases even having fallen over or, if remaining standing, often shedding their bark. The lesser increment in mortality among the untreated control steadily (and significantly; *p* < 0.01) increased from 1.0, 1.5, 7.8% and stabilised at 7.9% over the same period.

Clearer visualisation of the process of dieback development through the duration of the trial across all experiments can be seen in the stacked bar representation (Figure 5). While there was gradual, but low, background levels of dieback apparent in the untreated areas (control), which was capped at around 20%, disease progress in the treated sites showed clear acceleration. Thus, indicating a movement from class 5 downwards, resulting in high levels of mortality and almost negligible proportions of healthy plants remaining at 221 WAT.

### 2.3. Dieback Development in the South Infestation

Data collection for the south infestation was limited to the last two assessment dates beyond the initial point of treatment (0 WAT). Decline in tree health among the inoculated trees was significantly greater (*p* < 0.01) than that displayed by the untreated (control) sections (Figure 5 and Figure 6), falling to a health score of 1.40 and 0.67 at 165 and 221 WAT, respectively, in a trend similar to that seen in the north infestation (Figure 2).

Mortality assessment (Figure 7) of the south infection indicates a steady climb in plant death with 47.6 and 66.2% mortality at 165 and 221 WAT, respectively. Mortality rates among the untreated (control) plots were the same data as for the north infestation and significantly lower (*p* < 0.01) than for the treated section stabilising (7.8 and 7.9%) at the last two assessments.

### 2.4. South Infestation Transect

Overall plant health score across the 400 plants in the transect was low (Figure 5) at 1.19 compared to the untreated control at 4.36 (*p* < 0.01). Variability among assessment groups is illustrated by a heatmap diagram (Figure 8). Among these data, only one cluster of 10 plants was rated in top health (class 5). Overall mortality was determined at 49.2%, which was significantly greater (*p* < 0.01) than the untreated control (7.8%).

## 3. Materials and Methods

The location for this study was an isolated site identified as Corporal Dam (S 18°59′15″, E 137°39′27″) on Alexandria Station, a large cattle property (16,116 km^2^) in the Barkly Tablelands region of the Northern Territory, Australia (Figure 9). The vegetation type is best described as a Mitchell grass (*Astrebla*) tussock grassland [20], a tropical savannah bioregion typified by flat or rolling country dominated by C4 grasses and mostly lacking in trees. More specifically, it can be described as a Tropical Arid Grassland (MVG19-0) under the Australian major vegetation classification system [21]. The infestation of parkinsonia was located around a dam and its immediate catchment area. This is a typical habitat for this riparian weed as parkinsonia is generally associated with waterways and sites where cattle travel and congregate.

The study site consisted of two areas of reasonably dense and healthy parkinsonia. The smaller (1.9 ha) north infestation was adjacent to the dam (0.6 ha), with a population estimated to be 4000 plants. The larger (11.2 ha) south infestation was similarly densely covered with a population estimated at 24,000 trees (Figure 10).

The study commenced in August 2016. The north infestation was chosen for high-density inoculation with the bioherbicide Di-Bak Parkinsonia^®^ (BioHerbicides Australia Pty Ltd., Queensland, Australia), which at the time was in development, and as of December 2018, is registered through the Australian Pesticides and Veterinary Medicines Authority (APVMA; https://apvma.gov.au/ accessed on 10 May 2021). Di-Bak Parkinsonia is a formulation of three endophytic fungi from the phylum Ascomycetes (order Botryosphaeriales, family *Botryosphaeriaceae*), *Lasiodiplodia pseudotheobromae* A.J.L. Phillips, A. Alves & Crous (strain NT039), *Macrophomina phaseolina* (Tassi) Goid. (strain NT094) and *Neoscytalidium novaehollandiae* Pavlic, T.I.Burgess & M.J.Wingf. (strain QLD003). The fungi have been cultured separately on autoclaved white millet (*Panicum miliaceum* L.), dehydrated, combined and placed into size 0 hypromellose pharmaceutical-grade capsules (filled weight approximately 0.32 g). The south infestation was selected for limited treatment with the same bioherbicide to examine the potential for lateral movement of dieback from treated areas to adjacent plants. At commencement of the study, the overall health of plants at the site was observed to be excellent. Close observation through the subsequent treatment process (1785 trees) and extensive examination (more than 5000 trees) confirmed the site to be free of dieback (Figure 11a,b).

Inoculation of trees was achieved with the use of a prototype applicator device operated manually and powered by a cordless drill (Figure 12a). The applicator utilises an 8 mm diameter drill bit to create a 25 mm deep hole in the tree stem (at a height of 10–30 cm above soil level) located by sharp points pushed into the stem while the operator maintains forward pressure on the applicator against the tree. The action of withdrawing the drill fully primes the device with a single capsule (21.6 × 7.6 mm), and a sealing plug (15 × 7.5 mm), which are in tandem within each of the 15 chambers in the magazine (Figure 12b). The capsule and plug are then rapidly inserted into the fresh drill hole with a light plunging motion achieved by pushing the cordless drill forward using the drill bit as a ram. The capsule is lightly compressed within the hole, with the plug-end clearly visible on the stem surface (Figure 12c). As the capsules contain bioherbicidal fungi, only a single dose is delivered to the plant, irrespective of its size, resulting in the development of a stem lesion within months of treatment (Figure 12d).

### 3.1. North Infestation

Treatment of the densely populated north infestation was conducted according to the following principles: Where trees were in relative proximity (less than 0.5 m apart), the plant with the largest stem was treated before moving away at least 1 m to repeat the process. Where plants were more widely spaced or completely isolated, each tree with a stem thick enough to implant with a capsule (at least 40 mm diameter) was treated. This approach allowed the operators to move effectively throughout the infestation, ensuring good coverage of the treatment while taking into consideration plant density and distribution. Measurement of the distribution of treatments was achieved by recording a GPS waypoint (Garmin GPSmap 62 s) at the point where a full magazine (15 doses) was initiated and at every subsequent magazine change. On completion of treatment, GPS was used to map the external perimeter of the whole treated area and calculate its area (1.9 ha). Of the estimated 4000 trees, 1080 (72 magazines) were implanted with bioherbicide capsules (Figure 13).

### 3.2. South Infestation

Treatment of the south infestation (Figure 14) was limited to the southernmost tip and restricted to the western side of the creek line, which terminated at the dam to the north. Given that the population in this site was of lower density, and trees were generally of a larger size than in the north infestation, the inoculation strategy employed here was to treat every tree in this limited area with a single implanted bioherbicide capsule. The intention was to establish a seat of dieback with the potential to move in the direction of water flow [6,7]. The infestation boundary was determined by GPS tracking as described above (yielding an area of 11.2 ha). In total, 705 trees were implanted, requiring the use of 47 magazines, with waypoints captured as previously described. The total parkinsonia population was estimated at 24,000 trees. Work rate ranged between 101 and 145 trees per hour for the equipment operators across both the north and south infestations.

### 3.3. Reference Areas

Two sectors approximating rectangles were selected within the eastern side of the creek line within the south infestation as untreated reference (control) areas (Figure 14) with critical waypoints captured by GPS. The size of these sectors was 4250 and 3730 m^2^, respectively, containing more than 800 trees each.

### 3.4. Rating System and Assessment Dates

Parkinsonia tree health was evaluated using a modification of a visual scoring system [6], as outlined in Table 1. Each tree was carefully assessed to determine health by estimation of the percentage of the tree structure (stem and branches) considered alive. Leaves (amount present and health) are not considered a valuable determinant of plant condition, as this species readily sheds them under conditions of water stress. Healthy parkinsonia have green stems and branches that transition to yellow (chlorotic), then to red (severely chlorotic), followed by brown and black (necrotic) pigmentation as dieback develops. As parkinsonia trees grow, the loss of some functional material (generally smaller lower juvenile branches) is observed in normal (non-dieback affected) plants. The rating class 5 is valuable in differentiating highly healthy trees with normal (non-disease related) levels of branch loss from those showing slight symptoms of dieback (class 4).

Assessment was conducted on the north infestation at the trial establishment (week 0). At 0 WAT, 600 trees were assessed by rating three groups of 200 (which were further divided into clusters of 20) to establish the baseline condition of the experimental site. This was achieved by walking through each assessment area (the north infestation—Figure 13, and the two reference sectors—Figure 14) in a random pattern using GPS to ensure that repeated assessment of individual trees did not occur. Navigation in the more densely populated sections was made difficult, but not impossible, due to the thorny and dense nature of the infestation.

The north infestation was assessed for changes in health score at each subsequent time point (45, 64, 165 and 221 WAT). For the first two assessments, this was limited only to 200 treated trees (10 clusters of 20 each) which could be identified by either the presence of the inoculation plug, or where the plug had been lost, by the presence of a drill hole. At 165 WAT, 960 (treated) trees were easily identified for assessment, whereas, by 221 WAT, widespread tree death, lodging and the coating of tree stems with mud from a previous flooding event made identification of treated trees (by plug presence or dill holes) problematic. For this last assessment, 1200 randomly selected trees (60 clusters of 20 each) across the whole treatment zone were scored, representing approximately 30% of all trees in this study area. The untreated reference sectors (Figure 14) were assessed at 45 and 64 WAT by selecting 100 trees (5 × 20) from each of the two sectors and then increasing this to 200 trees (10 × 20) per sector at 165 and 221 WAT.

Assessment of the south infestation was conducted at the last two data collection events only (165 and 221 WAT), by which time there was a significant presence of dieback. As the trees in this study area were large and mostly free-standing, with almost 100% treated (705 in total), the assessment was conducted only on those trees where an inoculation hole could be identified, equating to 700 trees and 600 trees, as some had died, fallen over and even washed away by stream flows.

### 3.5. South Infestation Transect

At 221 WAT, general observations revealed a significant swath of dieback-affected plants on the eastern flank of the south infestation in what was previously observed to be a healthy zone between the treated zone (consisting of 705 treated trees) and the dam to the north (Figure 14). A sampling assessment approximating a linear transect was conducted from the northern (dam) end towards the previously treated (and dieback affected) southern zone. A true linear transect was impracticable due to restrictions in navigating through the dense and clumped population of parkinsonia. The method employed was to record a GPS waypoint, assess (using the previously described method) 10 parkinsonia trees that could be seen in a circle from that position, walk south a minimum of 5 m, then repeat the process. In total, 40 assessments were made (400 trees) over a linear distance of 290 m.

### 3.6. Weather Data

Rainfall data collected by a weather station at the Gallipoli outstation, 28 km southeast of the trial site, was provided by the North Australian Pastoral Company (NAPCo, Brisbane, QLD, Australia). Temperature data was accessed from the nearest location (Camooweal Township, S 19°55′12″ E 138°7′12″) 114 km SE of Corporal Dam [19].

### 3.7. Statistical Methods

Data analysis was performed with RStudio (V 4.1.0), using linear mixed models with post-hoc testing, and a comparison of means was conducted by Tukey’s HSD test at 0.01 confidence level. This approach considered the variations in sample size, which increased as the trial progressed and the categorical nature of the rating scales.

## 4. Discussion

Fungi in the family *Botryosphaeriaceae*, including those used in this study, are known for their ability to act as saprophytes, endophytic colonisers or latent pathogens [25]. For example, *M. phaseolina* (used in this study) is noted as one of the most destructive necrotrophic pathogens [26] due to its ability to produce a large repertoire of hydrolytic enzymes capable of degrading all major components of the plant cell wall and cuticle, facilitating the infection process. Furthermore, this species, which is favoured by enhanced activity at higher temperatures (30–35 °C), produces an extensive array of enzymes and toxins that disarm host defence mechanisms causing cell death and tissue degradation [26], accelerating the process of colonisation, tissue necrosis and plant death. More broadly, fungi in this family are well understood to initially colonise plants as endophytes, often infecting through natural openings, and then upon the onset of biotic and or abiotic stress, transition to a pathogenic strategy [25]. An examination of the climatic zones in which various Botryosphaeriaceae species were found to cause disease [27] assigned *M. phaseolina* to regions with a hot, humid summer, while *L. pseudotheobromae* (also used in this study) was assigned to regions with a warm, humid summer. The study site is considered transitional between hot, dry summer, mild winter, and hot, humid summer. An examination of the *Botryosphaeriaceae* found that 90% of the species in mango orchards in the Kimberley region (Western Australia) could be isolated from adjacent native vegetation [27] indicating the ubiquitous nature of these fungi as endophytes in northern Australia.

The results of this study clearly demonstrate that treatment of parkinsonia with stem-implanted bioherbicide capsules resulted in successful infection, colonisation and movement of dieback through a population; thereby, resulting in plant mortality and apparent suppression of re-colonisation from the (soil) seed bank. Previous research by [6,7,17,18] clearly demonstrated the ability of the fungi used in this study to reduce seed bank viability in laboratory and glasshouse studies. Meanwhile, field studies of naturally occurring dieback events showed that recruitment was prevented [6,7]. Elsewhere, field observations of parkinsonia sites indicate that seedling recruitment readily occurs beneath trees killed by herbicide treatment, but not generally at sites where dieback occurs. Ideal conditions for seedling recruitment occur in hot wet months after rain events or flooding [5], as experienced at this research site. Recruitment (which was not observed in this study) would not be expected in the healthy (control) areas due to adult plant competition (shading effects and possibly allelopathy).

The fungal isolates used in this study originated from locations [17] with hot, humid summers [27] and from the same climatic zone as the study site, and therefore would be considered well adapted to that temperature band.

The colonisation of treated trees was apparent at the first assessment point (45 WAT), with stem lesions clearly apparent, a highly significant decrease in health score and a small but significant increase in plant mortality. Health scores continued to fall, and tree mortality continued to increase over the length of the trial, while untreated controls remained relatively healthy over the same period of 221 weeks (4 years and 3 months). The overall outcome could only be described as an effective establishment (and ongoing) dieback event in the 1.9 ha north infestation resulting in an overall 81% mortality rate. A full survey of the site on final assessment revealed no visible signs of seedling recruitment, supporting the evidence [6,7,17,18] that the isolates used in this study are effective in reducing seedling emergence by processes of pre- and post-emergent pathogenesis.

Treatment of the lower (up-stream) end of the south infestation similarly resulted in effective dieback development (98%) and substantial mortality (66%) by the conclusion of the trial. This treatment zone served as an inoculum source for those untreated plants further north (downstream), with colonization evident for the whole 290 m towards the catchment point (Corporal Dam). This effective movement of the dieback front supported earlier observations [6,7] and clearly indicates that where parkinsonia has access to water flows in riparian locations, this method of treatment is effective in downstream dispersal of bioherbicide-induced dieback. Furthermore, the ability of dieback to spread from plant to plant suggests that treatment levels (proportion of plants inoculated) may be significantly reduced, particularly where plants are densely populated. However, reducing treatment levels could arguably limit the initial rate of plant-to-plant movements. Further research would establish appropriate density related treatment levels for optimal management of parkinsonia by this method.

A key factor in the success of the delivery mechanism is the rapid process of wounding, implantation and sealing the inoculum capsule into the stem of the target plant. Access to moisture from the xylem and phloem fluids and stem tissues, and the exclusion of an oxidizing and dehydrating atmosphere to the wound tissues, facilitates the absorption of moisture by the capsule contents activating the fungal agents, which then colonise the plant stem. As this is a process of inoculation by aggressive endophyte/pathogens, which are known to be favoured by wound entry [25], treatment success rates are understandably high. Furthermore, experimental observations elsewhere have shown that the process is not dose-dependent, with large trees succumbing to inoculation with the application of a single dose. The lightweight capsules are an efficient and highly portable delivery system, and freedom from using liquid-based herbicides greatly improves the ability of an operator to rapidly treat a parkinsonia infestation compared to conventional methods. 

Host range testing of this bioherbicide product was required to achieve registration through the APVMA. In those studies (BioHerbicides Australia un-published commercial-in-confidence data), 23 relevant native Australian woody plant species (along with *Parkinsonia aculeata*) were germinated and grown in ultra-high soil-applied doses of the bioherbicide (equivalent to 470 and 932 capsules m^2^). Among the species tested, only two (*P. aculeata* and *Eucalyptus microcarpa*) displayed reduced seedling emergence under greenhouse conditions. The bioherbicide was determined by the APVMA to be environmentally safe under its intended use in rangeland environments. The possibility of risk to un-intended species, such as *Mangifera indica* and *Persea americana,* is further reduced by the remoteness of the majority of parkinsonia infestations from agricultural cropping systems and the embargo on its use within specified distances from potentially susceptible species or production areas. These considerations greatly reduce, if not eliminate, the possibility of harming non-target species through environmental contamination.

A previous report on glasshouse experiments with parkinsonia failed to demonstrate the ability of endophytes isolated from diseased and healthy specimens to induce dieback symptoms [16]. The authors suggest that inoculation alone may not be sufficient to cause disease in the field. The research reported here shows that inoculation under field conditions can successfully cause parkinsonia dieback in the field and that it is capable of movement to untreated healthy plants, both adjacent to and a substantial distance away from the point of inoculation, thereby supporting observations elsewhere [6,7] in naturally occurring study locations. Inducing dieback in glasshouse-grown parkinsonia plants is problematic [16], with the absence of significant stress being indicated as a potential key factor in this process. A major source of stress is environmental temperature, and the conditions (25 °C) used by the authors [16] are significantly lower than those where fungi (such as *M. phaseolina* [26]) are most active (30–35 °C). Monthly mean maximum temperatures observed near the study site for this research ranged between 25.7 and 42 °C, supporting the observation that under much hotter conditions as normally experienced across much of northern Australia, inoculation of parkinsonia with a stem-implanted bioherbicide capsule is an effective approach for initiating a dieback event, leading to successful management of parkinsonia.

## 5. Conclusions

This is the first report of the use of a capsule-formulated bioherbicide delivered by a mechanized stem implantation device to successfully bring about environmental-scale control of *Parkinsonia aculeata*. The compact, robust, and convenient formulation of the bioherbicide as a capsule, and the rapid and easy delivery process, improves efficiency over other methods. The transmission of induced dieback through the plant population, killing both adult trees and preventing recruitment from the seed bank, creates a viable and environmentally appropriate alternative to resource-intensive physical and chemical control systems. 

## Figures and Tables

**Figure 1 plants-10-01909-f001:**
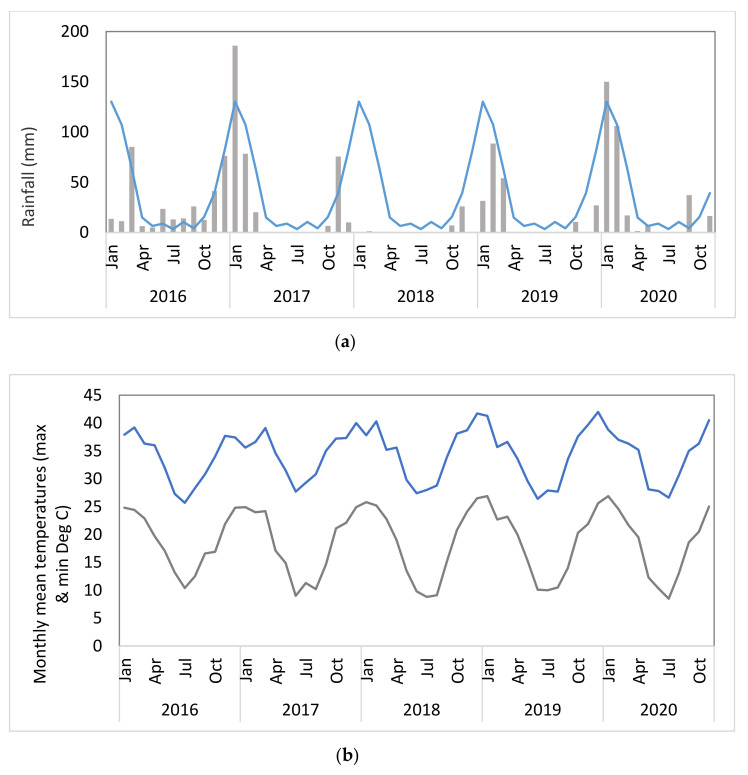
(**a**) Monthly rainfall totals (bars) recorded at Gallipoli outstation from January 2016 to October 2019. The line is indicative of the ten-year (2000–2019) monthly rainfall means. Data source, NAPCo. (**b**) Monthly mean maximum and mean minimum temperatures from Camooweal weather station [19].

**Figure 2 plants-10-01909-f002:**
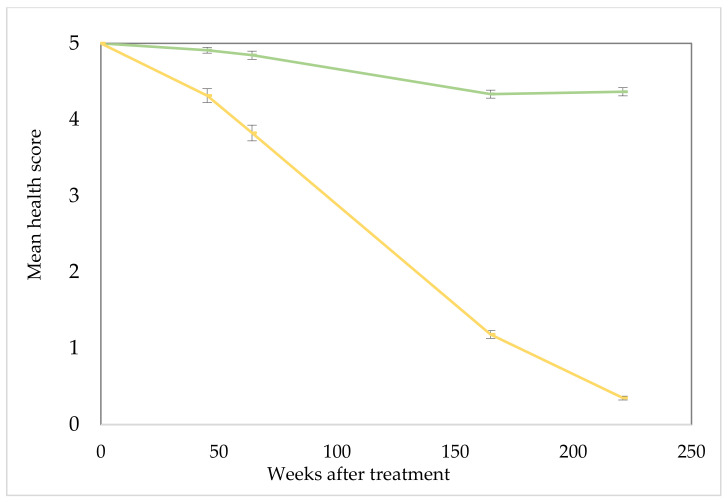
North parkinsonia infestation overall health score over trial duration (0, 45, 64, 165 and 221 WAT). Treated plants (gold), untreated control plants (green). Health score ratings, 5 = highest health score (>95% alive); 4 (71–95% alive); 3 (51–70% alive); 2 (31–50% alive); 1 (<30% alive); 0 (tree assessed as dead). Error bars represent ± standard error of means.

**Figure 3 plants-10-01909-f003:**
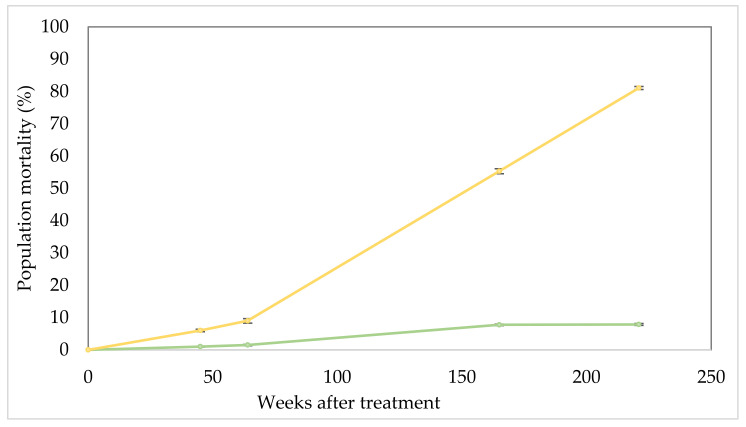
North parkinsonia infestation mortality over trial duration (0, 45, 64, 165 and 221 WAT). Treated plants (gold), untreated control plants (green). Error bars represent ± standard error of means.

**Figure 4 plants-10-01909-f004:**
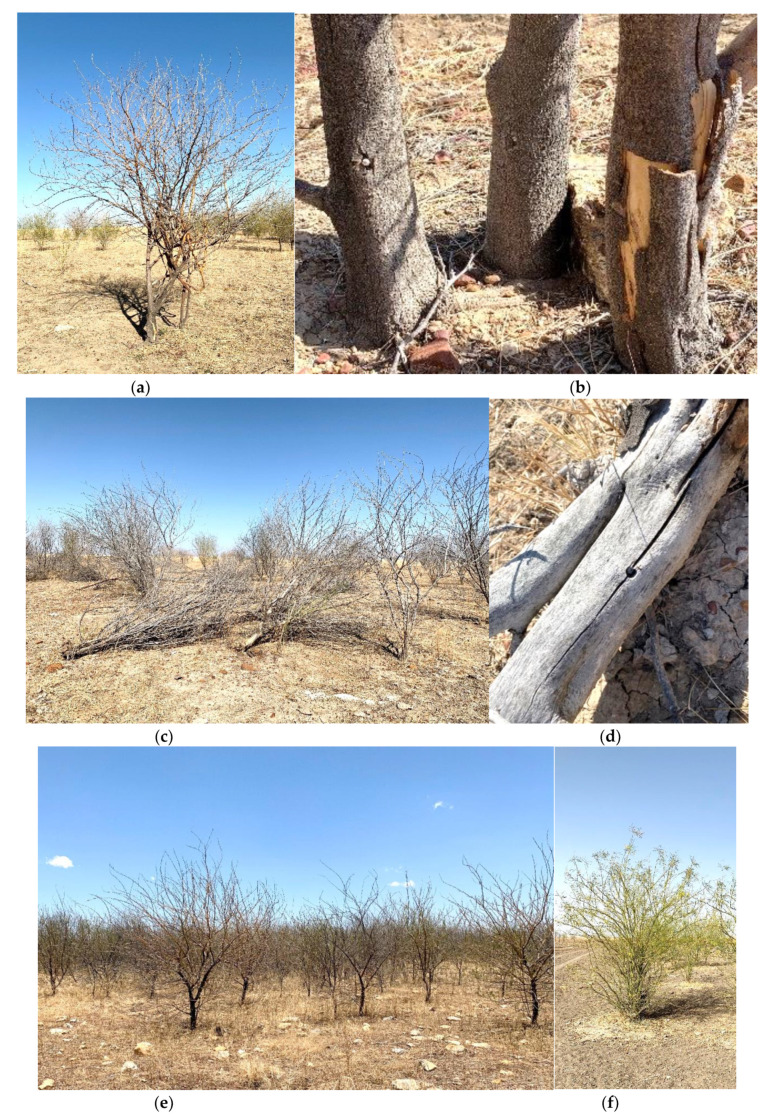
Result of parkinsonia treatment with implanted bioherbicide capsules at 221 weeks after treatment. (**a**) Free standing dead parkinsonia tree; (**b**) three individually treated stems with bark shedding evident on right specimen; (**c**) fallen dead specimens in north infestation; (**d**) stem of fallen dead specimen with inoculation hole evident; (**e**) effective and widespread mortality in treated area of south infestation; (**f**) live, healthy tree in untreated (control) area.

**Figure 5 plants-10-01909-f005:**
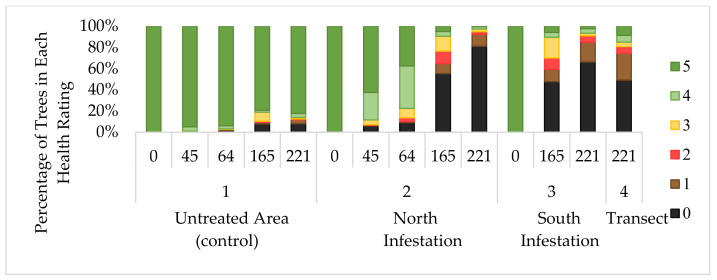
Progress of parkinsonia dieback represented as percentage distribution of each health rating class across control (1), north (2) and south infestations (3) over time (WAT) with comparison to transect (4) assessment. Ratings, 5 = highest health score (>95% alive); 4 (71–95% alive); 3 (51–70% alive); 2 (31–50% alive); 1 (<30% alive); 0 (tree assessed as dead).

**Figure 6 plants-10-01909-f006:**
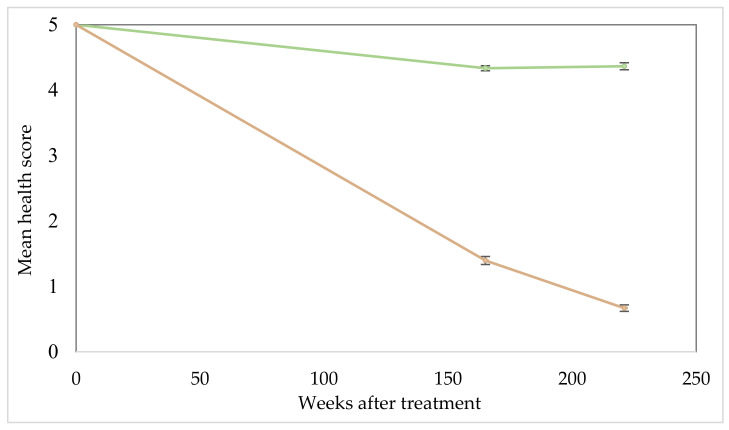
South parkinsonia infestation overall health score over trial duration (0, 165 and 221 WAT). Treated plants (gold), untreated control plants (green). Health score ratings, 5 = highest health score (>95% alive); 4 (71–95% alive); 3 (51–70% alive); 2 (31–50% alive); 1 (<30% alive); 0 (tree assessed as dead). Error bars represent ± standard error of means.

**Figure 7 plants-10-01909-f007:**
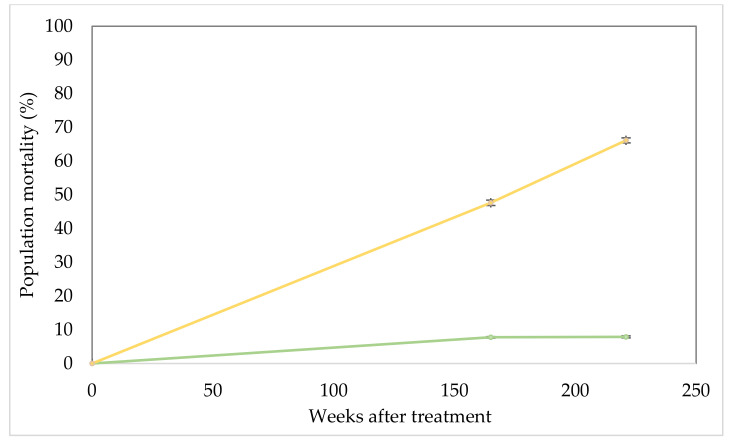
South parkinsonia infestation mortality over trial duration (0, 165 and 221 WAT). Treated plants (gold), untreated control plants (green). Error bars represent ± standard error of means.

**Figure 8 plants-10-01909-f008:**
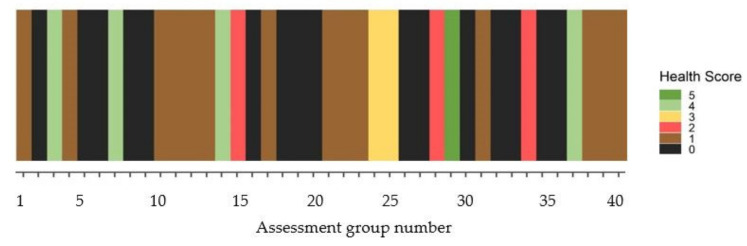
Heatmap of south infestation transect assessment points. Group 1 was located at the northernmost point, Group 40 closest to the treated area to the south. Health score ratings, 5 = highest health score (>95% alive); 4 (71–95% alive); 3 (51–70% alive); 2 (31–50% alive); 1 (<30% alive); 0 (tree assessed as dead).

**Figure 9 plants-10-01909-f009:**
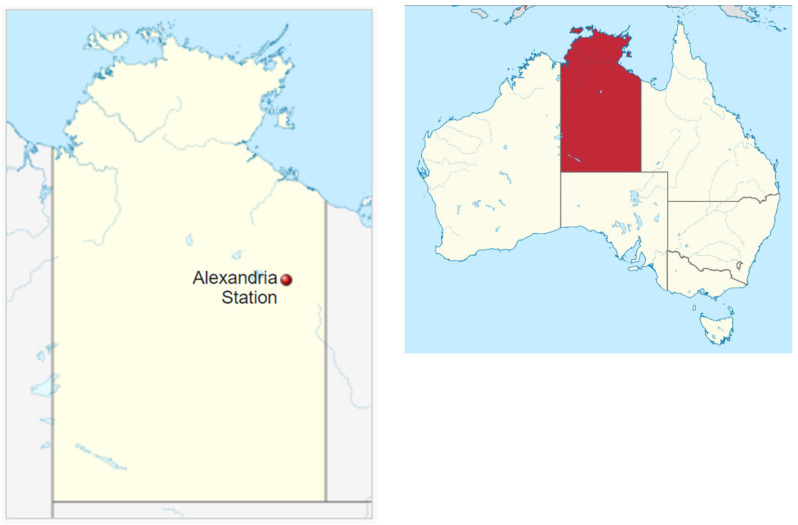
Location of Alexandria Station in the Northern Territory and location of the Northern Territory (shaded red inset) within Australia [22].

**Figure 10 plants-10-01909-f010:**
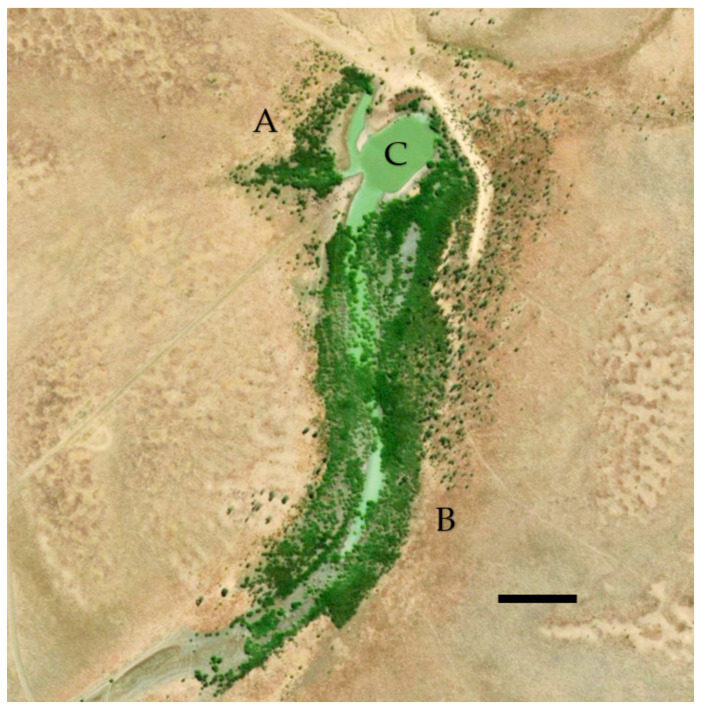
Satellite image of Corporal Dam study site with north infestation (**A**), south infestation (**B**) and dam (**C**). Vegetation almost entirely consisted of *Parkinsonia aculeata*. Scale bar = 100 m. Image [23] dated 2017.

**Figure 11 plants-10-01909-f011:**
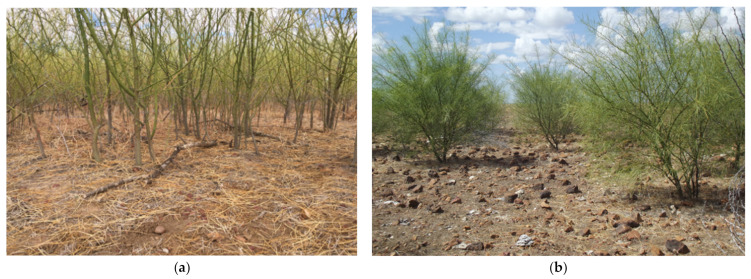
(**a**) Densely crowded healthy parkinsonia plants in north infestation; (**b**) free-standing healthy parkinsonia in south infestation at trial commencement.

**Figure 12 plants-10-01909-f012:**
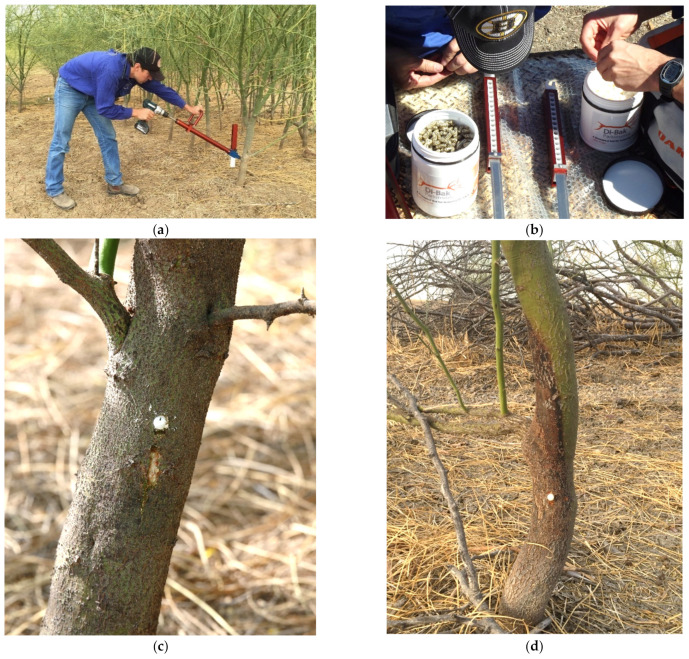
(**a**) Implanting a bioherbicide capsule into the lower stem of parkinsonia using a mechanical applicator; (**b**) loading the magazine with bioherbicide capsules and sealing plugs; (**c**) treated stem with sealing plug partially protruding from treatment hole; (**d**) parkinsonia stem six months after treatment showing a visible stem lesion in proximity to the treatment site.

**Figure 13 plants-10-01909-f013:**
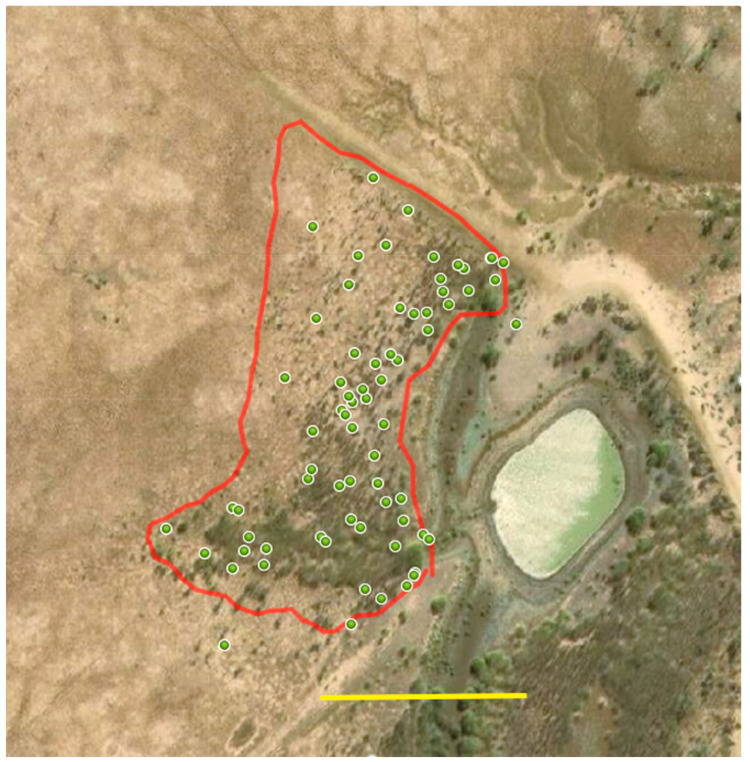
Outline (in red) of north infestation boundary as tracked by GPS surrounding the treatment area (1.9 ha); green markers indicating waypoints at which magazines of 15 doses were initiated. Scale bar = 100 m. Background imagery [24] accessed April 2021.

**Figure 14 plants-10-01909-f014:**
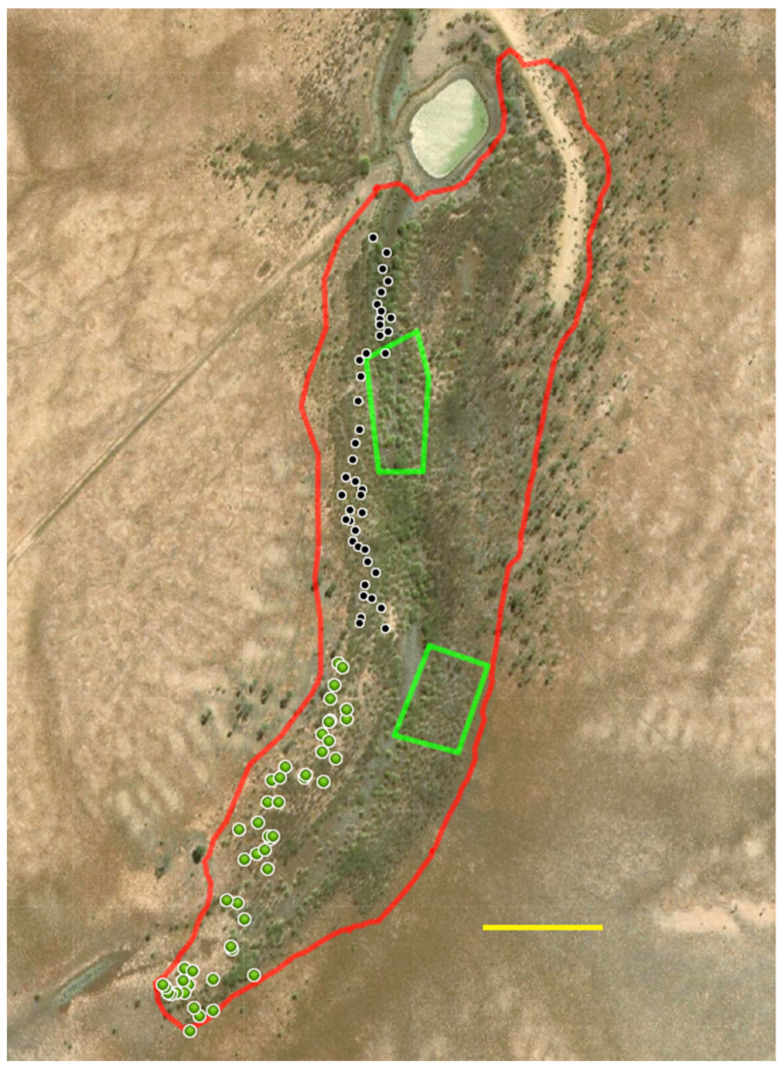
Outline (in red) of south infestation boundary as tracked by GPS (11.2 ha); green circle markers indicating waypoints at which magazines of 15 doses were initiated. Light green boundaries indicate reference (control) sectors. Black circle markers indicate assessment points for linear transect. Scale bar = 100 m. Background imagery [24] accessed April 2021.

**Table 1 plants-10-01909-t001:** Modified tree health scoring system for parkinsonia under field conditions.

Numeric Rating	Description of Tree Condition
5	>95% of tree remaining green and alive
4	71–95% of tree remaining green and alive
3	51–70% of tree remaining green and alive
2	31–50% of tree remaining green and alive
1	1–30% of tree remaining green and alive
0	Dead tree

## Data Availability

Not applicable.

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
