# Peer review of "Use of Stem Implanted Bioherbicide Capsules to Manage an Infestation of Parkinsonia aculeata in Northern Australia"

_plants, 2021, doi:10.3390/plants10091909_

Round 1

Reviewer 1 Report

This is a well-written and interesting article about the control of stablished plant pests using encapsulated biological agents. Moreover, I have to congratulate author for the long duration of this study. After reviewing this manuscript I recommend this paper to be published after major corrections mainly based on the discussión section, as it needs to be enriched with more references from previous studies of these or other fungi controlling weed pests.

Another aspect I think should be adressed in the discussion is the selectivity of these fungi affecting other vegetal species. In this case, vegetation was mainly compossed of Parkinsonia, but since these fungi are able to disperse across water flow, would they affect other plants in case there were present?

Some minor corrections are:

  • Correct the number of figures in titles and text
  • Indicated what is WAT the first time it is mentioned in the text, as m&m comes later than results
  • Correct the title axis of figures 12 and 13 since they are overhead with the numbers
  • Reformulate the sentence in lines 339-342, since it is so repetitive

Author Response

I thank the reviewer for these constructive comments.

I have addressed these suggestions in the manuscript, and furthermore, I have reviewed my text throughout making several additional changes to improve clarity and readability. 

Additional references have been included to support the research outcomes. 

Reviewer 2 Report

The research is well-written and of significant importance to the field. I recomment the publication of the work without hesitation.

Author Response

I thank the reviewer for their recommendation - I have made several adjustments to the manuscript to improve clarity.

Reviewer 3 Report

I found this to be a very interesting paper that was well-written and has major implications for future biological control work.  I had very few concerns with it, but a few minor issues are listed below.

On Lines 351-354, the point was made that no seedling recruitment was found, suggesting that the disease was affecting seedling establishment.  But were seedlings found to be establishing in untreated areas?  I don’t remember this being mentioned if it was.

On Lines 380-382, it was stated that this mixture of fungal organisms gives “species specificity” and eliminates the possibility of harming non-target species.  But is this really something that can be stated as a result of this work?  I am sure that with three different organisms at work that there must be other species that would be affected if this was attempted at sites where other tree species were growing.  For example, on Lines 336 to 338, it suggested these organisms could be found in mango orchards and native vegetation, though I realise this wasn’t specifically addressing the organisms used here.

My other comments really relate just to minor typographical issues.  The two sentences on Lines 27 to 31 appear to repeat each other and can be tidied up.

The SI convention of reporting metric units is to leave a space between the number and the unit yet there are many places in the paper where a space wasn’t left, especially in Lines 204 to 209 and Line 358.

Apart from that, brackets need closing at the end of the sentence finishing on Line 87.  The word “though” on Line 330 should presumably be “through.”  The word “untreated” on Line 357 doesn’t need a hyphen. 

Author Response

I thank the reviewer for thier detailed coments.  All recommendations have been addressed.

I have clarified the issue of seedling recruitment with a throrough explanation in the side margin and the additona of further information within the manuscript.

The issue of potential harm to non-target species is addressed by inclusion of reference to the biosafety work that was done in developing this product and supporting strategies used to minimise harm to agricultural systems.

Minor corrections have also been addressed and further adjustments were made to the manuscript to increase clarity.

Reviewer 4 Report

Accept in present form

Author Response

I thank the reviewer for reading this manuscript

Reviewer 5 Report

The manuscript is written very well. Material and Methods part are detailed. Statistical analysis is obust. Conclusions are interesting and relate strictly to the results of this study.

I have only minor suggestions:

The numbering of figures in the text should be in order, from 1 (presently 7) etc.

Figure 8 (should be 2), 9, 12 and 14 please, give the number of replicates in the figure description. Describe under the Figures, what does WAT mean?

The description of health ranking should be under all Figures that contain this ranking, etc. 8 and 12. Please change the Figure numbering, starting from 1, etc.

Please check if the order of chapter Material and Methods and Discussion is according to the journal's requirements.

Author Response

I thank the reviewer for their constructive comments.  The manuscrit has been re organised to meet the standard convention for this journal.  All figures have been re-numbered to reflect their changed location.

Other suggestions have been addressed.

Round 2

Reviewer 1 Report

Author has improved substantially the quality of the introduction and discussion as well as corrected some minor errors in the figures, so I propose this manuscript to be accepted in the present form.